# A Study on the Psychometric Properties of the Short Version of the Physical Activity Enjoyment Scale in an Adult Population

**DOI:** 10.3390/ijerph192215294

**Published:** 2022-11-19

**Authors:** Julian Fritsch, Susanne Weyland, Katharina Feil, Alexander Burchartz, Steffen Schmidt, Alexander Woll, Ulrich Strauch, Benjamin Wienke, Darko Jekauc

**Affiliations:** 1Institute of Sports and Sports Science, Karlsruhe Institute of Technology, 76131 Karlsruhe, Germany; 2Department of Sport and Exercise Psychology, Humboldt University of Berlin, 10117 Berlin, Germany

**Keywords:** dual-process, intrinsic motivation, maintenance, validity, reliability, gender invariance

## Abstract

A new measure of the short form of the Physical Activity Enjoyment Scale (PACES-S), including four items, has been developed that focuses on the subjective experience of enjoyment. As validation has so far only been conducted in a youth population, the purpose of the present article was to test the psychometric properties of the measure in an adult population in three studies. In the first study (*n* = 1017) the results supported the unidimensional structure of the instrument (χ^2^ = 10.0; df = 2; *p* < 0.01; CFI = 0.992; RMSEA = 0.063), revealed a satisfactory level of internal consistency (*ω* = 0.79), and showed that the measure is invariant across gender. The results on factorial validity and internal consistency were generally supported by the second study (*n* = 482), which additionally showed satisfactory test–retest reliability (*r* = 0.73). Finally, the third study (*n* = 1336) also supported the factorial validity and internal consistency of the measure and additionally showed a positive correlation with physical activity (*r* = 0.40), thus supporting the criterion-related validity of the measure. This more economical version of PACES seems to be particularly useful for large-scale studies.

## 1. Introduction

Physical activity is related to various indicators of physical and mental health. There is considerable research showing that higher levels of physical activity reduce the risk of cardiovascular diseases [1], different forms of cancer [2], depression [3], Alzheimer’s disease [4], and all-cause mortality [5]. Despite the evidence pointing clearly to the benefits of physical activity, many individuals do not meet common physical activity guidelines, such as to accumulate at least 150 min of moderate-intensity or 75 min of vigorous-intensity physical activity per week (for worldwide trends see [6]). Importantly, even if people manage to increase their physical activity, they often fail to sustain this increase over a longer period of time [7]. This discrepancy between the known benefits of regular physical activity on the one hand, and the failure of many people to maintain adequate levels of physical activity on the other hand, highlights the importance of understanding the psychological processes involved in maintaining physical activity.

It has been argued that enjoyment associated with feelings of fun and pleasure is an important psychological construct related to the maintenance of physical activity [8]. The basic idea that people who enjoy physical activity are also more active is in line with theoretical dual-process approaches that advocate the relevance of considering affective constructs alongside cognitive constructs (e.g., intention and self-efficacy) traditionally looked at in exercise psychology [9,10]. This theoretical reasoning is supported by the results of a meta-analysis, which included 114 independent samples and assessed enjoyment jointly with other related affective constructs (e.g., affective attitude), showing a positive relationship (*r* = 0.42) between these affective constructs and physical activity [11]. Moreover, in line with self-determination theory, the enjoyment that accompanies a behavior is viewed as an important information for an individual’s type of motivation [12]. Accordingly, physical activity that is perceived as more enjoyable leads to higher intrinsic motivation, which in turn increases the likelihood that the behavior will be maintained [13]. Considering the importance of developing a habit for long-term behavior change [14], it is also interesting to note that positive affective responses associated with enjoyment can contribute to the formation of habits [15,16]. In light of the importance of enjoyment for the maintenance of physical activity, the present article focuses on a new version of the Physical Activity Enjoyment Scale (PACES [8]) as likely the most prominent measure for enjoyment in exercise psychology. 

The original 18 item version of PACES was developed by Kendzierski and DeCarlo [8] and intended to be unidimensional. However, because this assumed unidimensionality was questioned by following confirmatory analyses [17], the original form has already undergone several forms of adaptations (see examples in [18,19,20,21]). Other than methodological problems of the shared variance of negatively and positively worded items (e.g., [18,22]), it was also assumed that some of the items (e.g., “Physical activity gives me energy”) confound the experience of enjoyment with its antecedents or consequences (e.g., [17,20]). For this reason, Chen, Weyland, Fritsch, Woll, Niessner, Burchartz, Schmidt and Jekauc [23] developed a questionnaire that focuses explicitly on the subjective experience of enjoyment. The psychometric analyses in this validation study, which focused on children and adolescents aged 11 to 17 years, revealed good internal consistency and test–retest reliability, supported the single-factor structure, and revealed a positive correlation with physical activity. The authors argued that this more economic version of PACES may be of particular use in large-scale studies [23].

### The Present Study 

The validation study on PACES-S focused on a youth population [23]. As the understanding of an instrument may differ between different groups of people [24], the purpose of the present article was to test the psychometric properties of PACES-S in an adult population. Another reason for the additional validation of PACES-S in an adult population lies in the finding that emotional processes were shown to vary across the life span [25]. Moreover, it should be noted that motives for physical activity were also shown to differ between adolescents and adults [26]. Thus, when PACES-S is intended to be used in future studies to examine the role of enjoyment in physical activity maintenance in adult populations, it is important to also establish its psychometric properties in this age group. This article consists of three studies in total. The first two studies were conducted in sport-related contexts. Factorial validity, internal consistency, and gender invariance were tested in the first study. In the second study, the results of factorial validity and internal consistency were cross-validated, with additional assessment of the test–retest reliability. Finally, in a third study, in addition to validating the findings on factorial validity and internal consistency in a more representative sample, criterion-related validity was examined by investigating the extent to which the instrument is related to physical activity. 

## 2. Material and Methods Study 1

### 2.1. Sample Study 1

Participants were recruited through university courses, fitness gyms, or sports clubs. There were no criteria for participants to take part in the study other than (a) being at least 18 years old and (b) being fluent in German. In total, 1017 participants (497 female, 2 missing data) with a mean age of 29.77 years (*SD* = 13.54, range = 18–83) took part in the study. 

### 2.2. Measure Study 1

#### PACES-S

The instrument PACES-S, including four items (i.e., “I enjoy it”, “I find it pleasurable”, “It is very pleasant”, “It feels good”) was used. The items were in German, were answered on a five-point Likert scale ranging from (1) strongly disagree to (5) strongly agree, and higher values reflect greater levels of enjoyment. A previous study has supported the psychometric properties of the German PACES-S in a youth population [23]. In terms of reliability, internal consistency ranged from *α* = 0.82 to *α* = 0.88 and test–retest reliability was *r* = 0.76. Moreover, confirmatory factor analysis (CFA) supported its factorial validity (χ^2^ = 53.62, df = 2, *p* < 0.001; RMSEA = 0.073; CFI = 0.99), while positive correlations (*r* = 0.21–0.44) with measures of physical activity were shown for its criterion-related validity. This previous validation study was based on the items of the German version of the long version of PACES, which has been validated in adolescents [22] as well as in adults [27]. 

## 3. Statistical Analyses Study 1

### 3.1. Factorial Validity

To validate the unidimensional factor structure identified in the study by Chen, Weyland, Fritsch, Woll, Niessner, Burchartz, Schmidt and Jekauc [23], CFA with full-information maximum likelihood estimation was performed in AMOS 26. The maximum likelihood estimation allows for a non-biased assessment of missing data [28]. The model fit was assessed by *χ*^2^-statistic, with a non-significant *p*-value showing a good model fit [29]. In light of the high sensitivity of this test in large samples [30], we additionally used the comparative fit index (CFI) and the root mean square error of approximation (RMSEA). CFI shows the relative fit improvement by comparing the suggested model with the baseline model, with values between 0.90 and 0.95 indicating an acceptable model fit and values above 0.95 indicating a good model fit [30]. RMSEA indicates the discrepancy between the suggested model with optimally chosen parameter values and the population covariance matrix, with values lower than 0.05 indicating a good model fit and values between 0.05 and 0.08 indicating an acceptable model fit [31]. However, it should be noted that, in the case of a small number of degrees of freedom, RMSEA tends to underestimate the model fit [32].

### 3.2. Invariance for Gender 

Regarding the measurement invariance for gender, four nested models (Model A to Model D) were calculated using AMOS 26 [33]. In each successive model, the previous model restrictions and additional constraints were included [34]. Thus, Model A tested the equivalence of the structure, Model B tested the equivalence of factor loadings, Model C tested the equivalence of the measurement intercepts, and Model D tested the invariance of item uniqueness and correlations between uniqueness across gender. Each of the models were tested by *χ*^2^ difference tests. As the *χ*^2^ difference test is sensitive to large sample sizes, it is recommended to also use the difference in CFI [35]. Accordingly, a ΔCFI-value of ≤ 0.01 indicates that the null hypothesis of invariance should not be rejected [35]. 

### 3.3. Internal Consistency 

Internal consistency was assessed by calculating McDonald’s omega using the macro for SPSS provided by Hayes and Coutts [36]. This indicator of internal consistency was chosen instead of the commonly used Cronbach’s alpha because the latter is criticized for its implicit assumption of tau-equivalence, which is often not met in psychological measures [36].

## 4. Results Study 1

The descriptive statistics of the individual items and the overall scale are shown in Table 1. Missing data was less than 0.2% for each item. The Little’s MCAR test was not significant (*χ*^2^ = 12.1, *df* = 6, *p* = 0.059), thus suggesting that data were missing completely at random. 

### Psychometric Properties 

The assessment of McDonald’s omega (*ω* = 0.79) indicated an acceptable internal consistency of the instrument. In terms of the factorial validity of PACES-S, the model showed an acceptable to good fit to the data (χ^2^ = 10.0; *df* = 2; *p* < 0.01; CFI = 0.992; RMSEA = 0.063). The individual items explained the variance between *R*^2^ = 0.397 and *R*^2^ = 0.559 and all the items were significantly loaded on the latent factor between 0.630 and 0.747. Regarding gender invariance, as shown in Table 2, the *χ*^2^-difference test was significant for Model B and Model D. However, because the CFI did not increase more than 0.01 for any comparison, the results suggest that the instrument is invariant across gender. 

## 5. Discussion Study 1

Based on the CFI value, the analysis of the factorial validity seems to indicate a good model fit [30]. When interpreting the RMSEA that indicates an acceptable model fit [31], it is important to note that this index tends to underestimate the model fit when the model has a small number of degrees of freedom [32]. The results of the invariance assessment suggest that the questionnaire can be used invariant across gender. Moreover, concerning the reliability of the instrument, McDonald’s omega (*ω* = 0.79) was in an acceptable range. In addition to the cross-validation of the findings for factorial validity and internal consistency, the test–retest reliability was also tested in Study 2. 

## 6. Material and Methods Study 2

### 6.1. Sample Study 2

Participants were recruited through university courses, fitness gyms, or sports clubs. As in Study 1, there were no criteria for participants to take part in the study other than (a) being at least 18 years old and (b) being fluent in German. In total, 482 participants (228 female) with a mean age of 26.03 years (*SD* = 11.06; range = 18–69) took part in the study. From those, 59 (33 female) participants with a mean age of 24.98 years (*SD* = 9.42; range = 18–61) took part in a second assessment for the test–retest reliability. 

### 6.2. Measure Study 2

#### PACES-S

The questionnaire PACES-S wasused again and the items were answered on a five-point Likert scale ranging from (1) strongly disagree to (5) strongly agree with higher values reflecting greater levels of enjoyment. For the test–retest reliability examination, the second assessment took place seven days after the first one. 

## 7. Statistical Analyses Study 2

The statistical procedures were similar to those in Study 1. To test the factorial validity of the questionnaire, a CFA was performed with CFI and RMSEA used as fit indices. For CFI, values between 0.90 and 0.95 indicate an acceptable model fit and values above 0.95 indicate a good model fit [30]. With regards to RMSEA, values lower than 0.05 indicate a good model fit and values between 0.05 and 0.08 an acceptable model fit [31]. In Study 2, reliability was assessed with (a) McDonald’s omega as an index for internal consistency, and (b) test–retest reliability. Test–retest reliability was examined by calculating Pearson product–moment correlation between PACES-S at the first and second assessment. 

## 8. Results Study 2

Means and standard deviations of the individual items, as well as the overall scale for both assessments, are shown in Table 3. Missing data was less than 0.7% for each item at the first assessment. The Little’s MCAR test was not significant (*χ*^2^ = 2.6, *df* = 3, *p* = 0.464), thus suggesting that data were missing completely at random. Moreover, there were no missing data at the second assessment.

### Assessment of Psychometric Properties 

Concerning factorial validity, the model showed a mixed fit to the data (*χ*^2^ = 10.4; *df* = 2; *p* < 0.01; CFI = 0.984; RMSEA = 0.093). In addition, the individual items explained the variance between *R*^2^ = 0.351 and *R*^2^ = 0.668 and all the items significantly loaded on the latent factor between 0.593 and 0.817. Regarding reliability, the values for McDonald’s omega (*ω* = 0.78 at the first assessment; *ω* = 0.87 at the second assessment), as well as for test–retest reliability (*r* = 0.73), indicated an acceptable to good reliability of the instrument. 

## 9. Discussion Study 2

The results of Study 2 generally supported the psychometric properties of PACES-S for adults. Regarding factorial validity, the CFI of 0.984 indicated a good model fit [30]. In contrast to Study 1, however, the RMSEA of 0.093 was outside the range of what is considered acceptable [31]. However, it should again be noted that the RMSEA may underestimate the model fit when the model has only a small number of degrees of freedom [32]. Moreover, given the smaller sample size (*n* = 482) than in Study 1 (*n* = 1017), in which the RMSEA indicated a better fit, it is also important to consider that this effect is even stronger in smaller samples [32]. Concerning reliability, the indices of McDonald’s omega (*ω* = 0.78–0.87) and test–retest reliability (0.73) were in an acceptable to good range [37]. When looking at the descriptive results of Study 1 and Study 2, the high values for the individual items (*M* = 4.12–4.80) are noticeable. This finding can be attributed to the fact that participants were recruited from university sports courses, fitness gyms, or sports clubs and it seems plausible that individuals recruited from such locations are likely to enjoy physical activity. For this reason, one objective of Study 3 was to test whether the results are transferrable to a more representative sample. The second objective of Study 3 was to investigate the criterion-related validity of PACES-S for adults by examining its relationship with physical activity.

## 10. Material and Methods Study 3

### 10.1. Sample Study 3

The sample was based on the Motorik-Modul Longitudinal Study (MoMo; [38]) as a submodule of the German Health Interview and Examination Survey for Children and Adolescents (KiGGS) conducted by the Robert Koch Institute [39]. The MoMo-submodule aims to examine participants’ development of health, physical fitness, and physical activity as well as their psychological, social, and environmental determinants [38]. For example, to assess physical fitness, the participants took part in tests for endurance or strength. Moreover, to test the health of participants, physical examinations, as well as self-reported questionnaires, were used. In the present study, the focus was on the relationship between enjoyment as a psychological determinant of physical activity and physical activity. As part of the KIGGS-study, in the baseline assessment as a first step, 167 communities were selected in Germany in the years 2003 to 2006, by proportionately considering the level of urbanization and geographic distribution. Using an age-stratified procedure, in a second step, children and adolescents aged 0 to 17 were randomly drawn from official registers. To examine the longitudinal trajectories of participants, three additional assessment waves were conducted after the baseline assessment, between 2009 and 2012, between 2015 and 2017, and between 2018 and 2022. Notably, the data collection of the last assessment wave (2018–2022) partially took place prior and partially during the Covid-19 pandemic, with the different conditions having the potential to confound the analysis. For this reason, the present study was based on the adult participants of the assessment wave between 2015 and 2017. The sample in Study 3 consisted of 1336 participants (742 female) with a mean age of 22.30 (*SD* = 3.25, range = 18 to 31) years.

### 10.2. Measure Study 3

#### PACES-S

The questionnaire PACES-S [23] was again used, and as in the previous two studies, items were answered on a five-point Likert scale ranging from (1) strongly disagree to (5) strongly agree with higher values reflecting greater levels of enjoyment.

### 10.3. Physical Activity 

The MoMo Physical Activity Questionnaire was used to measure physical activity [40]. This questionnaire contains 28 items to measure physical activity in school, during leisure time, and in organized sports clubs. For those participants that were not at school anymore, physical activity at work was assessed. Previous findings indicate that this questionnaire has a moderate test–retest reliability (*r* = 0.68) and a moderate correlation (*r* = 0.29) with accelerometer-recorded data [40]. In the present study, the outcome measure was the amount in minutes of moderate to vigorous physical activity. 

## 11. Statistical Analyses Study 3

Regarding factorial validity, a CFA was again performed with CFI and RMSEA used as fit indices. For CFI, values between 0.90 and 0.95 indicate an acceptable model fit and values above 0.95 indicate a good model fit [30]. For RMSEA, values lower than 0.05 indicate a good model fit and values between 0.05 and 0.08 indicate an acceptable model fit [31]. For factorial validity, the same statistical procedures were conducted as in Study 1 and Study 2. Reliability was again assessed with McDonald’s omega. Moreover, for criterion-related validity, a correlation between physical activity enjoyment and weekly minutes of physical activity in organized sports clubs, leisure time, or at work measured by questionnaire was calculated. 

## 12. Results Study 3

Means and standard deviations of the individual items and the overall scale, as well as McDonald’s omega, are shown in Table 4. Missing data was less than 1.7% for each item. The Little’s MCAR test was not significant (*χ*^2^ = 6.6, *df* = 14, *p* = 0.951), thus suggesting that data were missing completely at random.

### Assessment of Psychometric Properties 

With regards to the factorial validity of PACES-S, the model showed an acceptable to good fit to the data (*χ*^2^ = 13.5; *df* = 2; *p* < 0.01; CFI = 0.996; RMSEA = 0.066). The individual items explained the variance between *R*^2^ = 0.615 and *R*^2^ = 0.727 and all items were significantly loaded on the latent factor between 0.784 and 0.853. In addition, McDonald’s omega (*ω* = 0.88) indicated a good internal consistency of the instrument. Concerning the criterion-related validity, the correlation between physical activity (*M* = 199.20 minutes per week; *SD* = 204.39) and PACES-S was significant (*r* = 0.40; *p* < 0.01). 

## 13. Discussion Study 3

Using a more representative sample than in the previous two studies, the results of Study 3 generally also supported the psychometric properties of PACES-S for adults. In addition, the findings showed a positive correlation between PACES-S for adults with physical activity in support of its criterion-related validity. 

## 14. General Discussion

The purpose of the present article was to test the psychometric properties of PACES-S [23] in adult populations using three different samples. Considering that emotional processes were shown to differ across age groups [25], testing the psychometric properties of PACES-S in an adult population is important to justify its use in future studies in this age group. Regarding factorial validity, the results generally indicate an acceptable to good model fit and that the instrument is invariant across gender. With regards to reliability, the results indicate, in most cases, good indices of internal consistency and a moderate test–retest reliability. Finally, concerning criterion-related validity, the results indicate a positive relationship with physical activity. Taken together, these findings indicate that PACES-S is also a valid and reliable instrument to assess enjoyment in an adult population. 

Looking at the fit indices of the confirmatory analyses, it is apparent that the CFI consistently indicated a good model fit [30], whereas the RMSEA shows an acceptable model fit in Study 1 and Study 3 and even a poor model fit in Study 2 [31]. This seemingly contradictory finding might be explained by simulations showing that the RMSEA tends to underestimate the model fit with a small number of freedoms [32]. For this reason, consistent with Chen, Weyland, Fritsch, Woll, Niessner, Burchartz, Schmidt and Jekauc [23], the results are generally supportive of the unidimensional structure of PACES-S for adults. This finding is of particular interest in light of the results questioning the unidimensionality of previous forms of PACES [17,18,27]. Here, the selection of items based on a definition that focused on the subjective experience of enjoyment in the study by Chen, Weyland, Fritsch, Woll, Niessner, Burchartz, Schmidt and Jekauc [23] may have helped to exclude items that focus on antecedents or consequences rather than the experience of enjoyment. Moreover, in addition to procedures used in the study on PACES-S in a youth population [23], the present study showed that PACES-S for adults can be used invariantly across gender. 

Regarding reliability, McDonald’s omega was between 0.78 and 0.88 in the different samples. These values are similar, although somewhat lower, compared to the validation study in a youth population [23]. The recruitment of the samples in Study 1 and Study 2 from sport-related contexts may have implied a restriction of range, which could explain why McDonald’s omega was the highest in the more representative sample in Study 3 [41]. Moreover, while in longer versions of PACES the internal consistency was often found to be above 0.90 (e.g., [8,25,39]), it is important to note that a low number of items, as in PACES-S for adults, negatively affects indicators of internal consistencies [42]. The test–retest reliability as an additional indicator was *r* = 0.73, which is similar to [23], and also similar to the results of a study on the long version of PACES [22]. Thus, based on common norms of reliability indices [37], the findings indicate an acceptable to good reliability of PACES-S for adults. 

The results further indicated that PACES-S for adults has a positive relationship with physical activity supporting its criterion-related validity. The correlation of this relationship (*r* = 0.40) is similar to those shown in the study by Chen, Weyland, Fritsch, Woll, Niessner, Burchartz, Schmidt and Jekauc [23] and also to meta-analytical findings on the relationship between affective constructs and physical activity in general [11]. This correlation is also in a similar range to the effect sizes of the relationship between other psychological constructs, such as intention or perceived behavioral control, and physical activity [43,44]. From a theoretical point of view, this finding therefore also supports dual-process approaches, which argue for considering affective constructs alongside cognitive constructs when explaining physical activity [9,10]. 

## 15. Strengths and Limitations

The large sample size (*n* = 2835) across the three studies is a strength of the present article. In this regard, the more representative sample based on the MoMo-study in Study 3 allowed us to replicate the findings from the first two studies, in which participants were recruited from sport-related contexts. Moreover, sophisticated statistical analyses involving structural equation modeling were used to assess the psychometric properties of the instrument. While the samples were restricted to adult participants, a limitation of this study is that the samples tended to consist of participants in a younger adult age group (mean age between 22.30 and 29.77). Generally, it would be important to also test the use of PACES-S in other contexts (e.g., in a clinical setting). Moreover, for criterion-related validity, the results were based solely on a subjective assessment of physical activity. While the use of questionnaires to assess physical activity is advantageous in large sample sizes [45], it seems worthwhile for future studies to complement such methods with device-based methods(e.g., accelerometer [46]). Moreover, the cross-sectional study design does not allow us to assess whether enjoyment can actually contribute to the maintenance of physical activity. Finally, as in the study by Chen, Weyland, Fritsch, Woll, Niessner, Burchartz, Schmidt and Jekauc [23], the results are restricted to German-speaking participants, thus pointing to the importance of also testing the psychometric properties in other languages. 

## 16. Conclusions

In conclusion, the findings of the present article suggest that the recently developed PACES-S reveals not only good psychometric properties in youth populations but can also be used in adult populations. This finding is important given that emotional processes might differ across age groups [25]. In particular, the results indicated satisfactory reliability and validity of the instrument, which is comparable to longer versions of PACES. This more economic version of PACES, focusing on the subjective experience of enjoyment, may be of particular use in large-scale studies. In this way, we hope that PACES-S for adults will help to clarify the role of enjoyment in the maintenance of physical activity and thus contribute to a better understanding of why some individuals maintain being active and others do not. 

## Figures and Tables

**Table 1 ijerph-19-15294-t001:** Descriptive statistics of Study 1.

	*M*	*SD*	*ω*
I enjoy physical activity.	4.71	0.55	
I find physical activity pleasurable.	4.37	0.75	
Physical activity is very pleasant.	4.20	0.83	
Physical activity feels good.	4.57	0.61	
PACES-S	4.46	0.53	0.79

Note. *M* = mean; *SD* = standard deviation; *ω* = McDonald’s omega.

**Table 2 ijerph-19-15294-t002:** Analysis of invariance across gender.

Model	*χ* ^2^	*df*	*p*	CFI	RMSEA	ΔCFI	Δ*χ*^2^	Δ*df*	*p*
Model A	12.607	4	<0.05	0.992	0.046				
Model B	25.433	7	<0.01	0.983	0.051	0.009	12.826	3	<0.01
Model C	26.589	10	<0.01	0.984	0.040	0.001	1.156	3	>0.05
Model D	37.757	14	<0.01	0.978	0.041	0.006	11.168	4	<0.05

Note. *χ*^2^ = chi-square; *df* = degrees of freedom; *p* = probability value; CFI = Comparative Fit Index; RMESA = Root Mean Square of Approximation; ΔCFI = difference in CFI; Δ*χ*^2^ = chi-square difference; Δ*df* = differences in degrees of freedom.

**Table 3 ijerph-19-15294-t003:** Descriptive statistics of Study 2.

	*M* _1_	*SD* _1_	*ω* _1_	*M* _2_	*SD* _2_	*ω* _2_
I enjoy physical activity.	4.80	0.52		4.76	0.50	
I find physical activity pleasurable.	4.37	0.67		4.42	0.62	
Physical activity is very pleasant.	4.12	0.72		4.22	0.77	
Physical activity feels good.	4.63	0.72		4.61	0.58	
PACES-S	4.48	0.49	0.78	4.50	0.52	0.87

Note. *M*_1_ = mean at first assessment; *SD*_1_ = standard deviation at first assessment; *ω*_1_ = McDonald’s omega at first assessment; *M*_2_ = mean at second assessment; *SD*_2_ = standard deviation at second assessment; *ω*_2_ = McDonald’s omega at second assessment.

**Table 4 ijerph-19-15294-t004:** Descriptive statistics of Study 3.

	*M*	*SD*	*ω*
I enjoy physical activity.	4.23	0.82	
I find physical activity pleasurable.	3.89	0.89	
Physical activity is very pleasant.	3.84	0.91	
Physical activity feels good.	4.19	0.83	
PACES-S	4.03	0.75	0.88

Note. *M* = mean*; SD* = standard deviation; *ω* = McDonald’s omega.

## Data Availability

The datasets generated and analyzed during the current study are not publicly available due to the strict ethical standards required by the Federal Office for the Protection of Data with which study investigators are obliged to comply but are available from the corresponding author on reasonable request.

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
