# Peer review of "A Study on the Psychometric Properties of the Short Version of the Physical Activity Enjoyment Scale in an Adult Population"

_ijerph, 2022, doi:10.3390/ijerph192215294_

Round 1

Reviewer 1 Report

Many thanks for having me to review the manuscript titled “A study on the psychometric properties of the short version of the physical activity enjoyment scale in an adult population”. The authors draw attention to the enjoyment of physical activity impacting the maintenance of physical activity for the adult population via the uni-dimensionality model. I appreciate the authors further testing the validation of PACES-S in the adult population and bringing the importance of enjoyment and the predictor of improving physical activity with a good and clear structure presentation. For this manuscript, I suggest the authors have a very tiny revision before moving toward publication. Please see my comments to the authors below. 

For the introduction, the authors provided adequate studies demonstrating the solid relationship between physical activity and individuals’ well-being, and the benefit of enjoyment in maintenance. If possible, authors may better explain the reasons of testify the PACES-S for adults, even if the validation in adolescent groups has been verified.

For the method section, the authors possibly better show a bit more details for those studies. In studies 1 and 2, participants were recruited from the activity contexts, while the maintenance seemed not to be verified, whether the level of maintenance has been overrated. Or authors conducted any additional standards to include the participants, if so, the authors needed to list them. As to study 3, authors may need to provide more information about the MoMo-submodule and its relationship with this study 3. In addition, the authors may better provide more explanation for the statistical approaches, why and how to select and fit into the analysis.

For discussion, authors may need to illustrate the meaning and innovation of the current study, comparing only generalising the PACES-S from adolescent to adult groups.   

Overall, I believe verifying and generalising psychometric properties are valuable and important. The Authors were well structured and carefully presented their ponderable results, but need a bit more explanation and illustration to publish level.

 Possible useful citations 

Richard Whittington, Charlotte Pollak, Alice Keski-Valkama, Andrew Brown, Alina Haines-Delmont, Jesper Bak, Jacob Hvidhjelm, Roger Almvik & Tom Palmstierna (2022) Unidimensionality of the Strengths and Vulnerabilities Scales in the Short-Term Assessment of Risk and Treatability (START), International Journal of Forensic Mental Health, 21:2, 175-184, DOI: 10.1080/14999013.2021.1953193

Shou, Y., Sellbom, M., & Chen, H. (2022). Fundamentals of Measurement in Clinical Psychology. Comprehensive Clinical Psychology (Second Edition), 13-35. https://doi.org/10.1016/B978-0-12-818697-8.00110-2

Author Response

For the introduction, the authors provided adequate studies demonstrating the solid relationship between physical activity and individuals’ well-being, and the benefit of enjoyment in maintenance. If possible, authors may better explain the reasons of testify the PACES-S for adults, even if the validation in adolescent groups has been verified.

We explain now in more detail why it is important to also testify PACES-S in an adult population.

Lines 104-113: The validation study on PACES-S focused on a youth population [1]. Because the understanding of an instrument may differ between different groups of people [24], the purpose of the present article was to test the psychometric properties of PACES-S in an adult population. Another reason for the additional validation of PACES-S in an adult population lies in the finding that emotional processes were shown to vary across the life span [25]. Moreover, it should be noted that also motives for physical activity were shown to differ between adolescents and adults [26]. Thus, when PACES-S is intended to be used in future studies to examine the role of enjoyment for physical activity maintenance in an adult population, it is important to establish its psychometric properties also in this age group.

For the method section, the authors possibly better show a bit more details for those studies. In studies 1 and 2, participants were recruited from the activity contexts, while the maintenance seemed not to be verified, whether the level of maintenance has been overrated. Or authors conducted any additional standards to include the participants, if so, the authors needed to list them. As to study 3, authors may need to provide more information about the MoMo-submodule and its relationship with this study 3. In addition, the authors may better provide more explanation for the statistical approaches, why and how to select and fit into the analysis.

For Study 1 and 2, there were no criteria to take part in the study other than being at least 18 years old and being fluent in German. We added this information when describing the sample.

Lines 123-125: There were no criteria for participants to take part in the study other than (a) being at least 18 years old and (b) being fluent in German.

Lines 206-208: As in Study 1, there were no criteria for participants to take part in the study other than (a) being at least 18 years old and (b) being fluent in German.

Regarding Study 3, we provide now more information about the MoMo-submodule and its relationship with Study 3.

Lines 266-274: The sample was based on the Motorik-Modul Longitudinal Study [MoMo; 35] as a submodule of the German Health Interview and Examination Survey for Children and Adolescents (KiGGS) conducted by the Robert Koch Institute [36]. The MoMo-submodule aims to examine participants’ development of health, physical fitness, and physical activity as well as their psychological, social, and environmental determinants [35]. For example, to assess physical fitness, the participants take part in tests for endurance or strength. Moreover, to test the health of participants, physical examinations as well as self-reported questionnaires are used. In the present study, the focus is on the relationship between enjoyment as a psychological determinant and physical activity.  

Moreover, we provide now more information when presenting the statistical analyses for Study 2 and Study 3.

Lines 219-227: The statistical procedures were similar to Study 1. To test the factorial validity of the questionnaire, a CFA was performed with CFI and RMSEA used as fit indices. For CFI, values between .90 and .95 indicate an acceptable model fit and values above .95 indicate a good model fit [27]. With regards to RMSEA, values lower than .05 indicate a good model fit and values between 0.05 and 0.08 an acceptable model fit [28]. In Study 2, reliability was assessed with (a) McDonald’s omega as an index for internal consistency and (b) test-retest reliability. Test-retest reliability was examined by calculating Pearson product-moment correlation between PACES-S at the first and second assessment.

Lines 300-308: Regarding factorial validity, a CFA was again performed with CFI and RMSEA used as fit indices. For CFI, values between .90 and .95 indicate an acceptable model fit and values above .95 indicate a good model fit [27]. For RMSEA, values lower than .05 indicate a good model fit and values between 0.05 and 0.08 an acceptable model fit [28]. Reliability was again assessed with (a) McDonald’s omega as an index for internal consistency. Moreover, for criterion-related validity, a correlation between physical activity enjoyment and weekly minutes of physical activity in organized sports clubs, leisure time, or at work measured by questionnaire was calculated.

For discussion, authors may need to illustrate the meaning and innovation of the current study, comparing only generalising the PACES-S from adolescent to adult groups.   

At the beginning of the discussion, we emphasize now the importance of having a valid and reliable instrument of PACES-S also for the adult population.

Lines 332-341: The purpose of the present article was to test the psychometric properties of PACES-S [1] in adult populations using three different samples. Considering that emotional processes were shown to differ across age groups [25], testing the psychometric properties of PACES-S in an adult population is important to justify its use in future studies in this age group. Regarding factorial validity, the results generally indicate an acceptable to good model fit and also that the instrument is invariant across gender. With regards to reliability, the results indicate in most cases good indices of internal consistency and a moderate test-retest reliability. Finally, concerning criterion-related validity, the results indicate a positive relationship with physical activity. Taken together, these findings indicate that PACES-S is also a valid and reliable instrument to assess enjoyment in an adult population.

Moreover, in the conclusion section of the article, we also stress more explicitly now the importance of having a valid and reliable instrument for adult populations.

Lines 398-407: In conclusion, the findings of the present article suggest that the recently developed PACES-S reveals not only good psychometric properties in youth populations, but can also be used in adult populations. This finding is important given that emotional processes might differ across age groups [25]. In particular, the results indicate satisfactory reliability and validity of the instrument, which is comparable to longer versions of PACES. This more economic version of PACES focusing on the subjective experience of enjoyment may be of particular use in large-scale studies. In this way, we hope that PACES-S for adults will help clarify the role of enjoyment in the maintenance of physical activity and thus contribute to a better understanding of why some individuals maintain being active and others do not.

Reviewer 2 Report

This article concerns one of the important factors of a healthy lifestyle, which is physical activity. The Authors undertook a validation study of the psychometric properties of the abbreviated form of the Physical Activity Enjoyment Scale (PACES-S) in the adult population. The article consists of three studies and is related to the involvement of the respondents in physical activity, which in a subjective opinion is a source of pleasure for them. The large sample size (n = 2,835) in all three studies is noticeable, which is a great advantage of the validation study.

Congratulates the Authors of this study, because on the basis of the validation process, which determines the accuracy and reliability of the research tool, which is the abbreviated version of this scale (PACES-S), it will be possible to use it in large population studies on the subjective assessment of satisfaction with undertaken physical activity in various social groups of people adults. The validation process is a tedious and time-consuming study. I have read the article carefully and in my opinion there are no aspects for substantive and technical improvement.

Detailed review of the article according to the attached points

1. What is the main question addressed by the research?

The aim of this article is to evaluate the psychometric properties of the abbreviated form of the Physical Activity Enjoyment Scale (PACES-S) in the adult population based on three studies. The PACES-S focuses on the subjective experience of pleasure in physical activity. So far, validation of this scale has been carried out only in the youth population.

2. Do you consider the topic original or relevant in the field? Does it address a specific gap in the field?

The subject of the article is of utmost importance for the area of public health, because it concerns one of the basic factors of a healthy lifestyle, which is physical activity. The innovative nature of this study results from the fact that the authors, for the first time, validated the PACES-S scale for the adult population. The results indicate a satisfactory reliability and validity of this scale. This analysis may be especially useful for other multicentre studies on physical activity. It can help explain the role of the joy of being physically active, and therefore contribute to a better understanding of why some people take up and continue physical activity and others don't.

3. What does it add to the subject area compared with other published material?

This study, conducted on a large group of adults (n = 2,835) in all three stages of the study, is a great advantage of the validation study. Although the majority of these people are young adults exposed to physical activity in the gym, so far there has been no validation of the PACES-S scale for other age groups except adolescents. The article consists of three studies. The first two studies were conducted in sport-related contexts. The first study examined factor validity, internal consistency, and gender invariability. In the second study, the results of factor validity and internal consistency were cross-validated with an additional assessment of test-retest reliability. In the third study, in addition to validating the results, the criteria validity was also assessed by examining the degree to which the tool is related to physical activity.

4. What specific improvements should the authors consider regarding the methodology? What further controls should be considered?

In my opinion, the validation study methodology used is correct. On the other hand, the studied group of adults could be more precisely characterized in terms of demographic and health factors. The sample of adults in future research could also come from people outside of gyms and sports clubs.

5. Are the conclusions consistent with the evidence and arguments presented and do they address the main question posed?

The conclusions are fully consistent with the evidence presented and the arguments carried out after each stage of the study and in the general discussion. The conclusions also relate to the main question posed for the purpose of the work.

6. Are the references appropriate?

All the references used in the article are included in the text of the work. These items are up-to-date and well-matched to the topic of the article and validation research.

7. Please include any additional comments on the tables and figures.

I have no comments to the text in the tables.

Author Response

  1. What specific improvements should the authors consider regarding the methodology? What further controls should be considered?

In my opinion, the validation study methodology used is correct. On the other hand, the studied group of adults could be more precisely characterized in terms of demographic and health factors. The sample of adults in future research could also come from people outside of gyms and sports clubs.

In the “Strength and limitations” section, we added that it is important in future research to also focus on other samples.

Lines 385-388: Although the samples were restricted to adult participants, a limitation of this study is that the samples tended to consist of participants in a younger adult age group (mean age between 22.30 and 29.77). Generally, it would be important to test the use of PACES-S also in other contexts (e.g., in a clinical setting).